# LABELFOOL:
# A TRICK IN THE LABEL SPACE

## ABSTRACT

It is widely known that well-designed perturbations can cause state-of-the-art machine learning classifiers to mis-label an image, with sufficiently small perturbations that are imperceptible to the human eyes. However, by detecting the inconsistency between the image and wrong label, the human observer would be alerted of the attack. In this paper, we aim to design attacks that not only make classifiers generate wrong labels, but also make the wrong labels imperceptible to human observers. To achieve this, we propose an algorithm called LabelFool which identifies a target label similar to the ground truth label and finds a perturbation of the image for this target label. We first find the target label for an input image by a probability model, then move the input in the feature space towards the target label. Subjective studies on ImageNet show that in the label space, our attack is much less recognizable by human observers, while objective experimental results on ImageNet show that we maintain similar performance in the image space as well as attack rates to state-of-the-art attack algorithms.

## 1 INTRODUCTION

Deep neural networks are powerful learning models that achieve state-of-the-art pattern recognition performance in classification tasks (Krizhevsky et al., 2012b; LeCun et al., 2010; He et al., 2016). Nevertheless, it is found that adding well-designed perturbations to original samples can make classifiers of deep neural networks fail (Szegedy et al., 2013). These kinds of samples are called *adversarial samples*. Techniques for generating adversarial samples are called attackers.

We think the ideal attacker should satisfy three levels of requirements. The first requirement is fooling networks which means making classifiers fail to classify an image successfully. For example, a dog image can be classified as a cat after added some well-designed perturbations. There are a number of methods for achieving a high attack rate (Goodfellow et al., 2015; Carlini & Wagner, 2017; Dong et al., 2018).

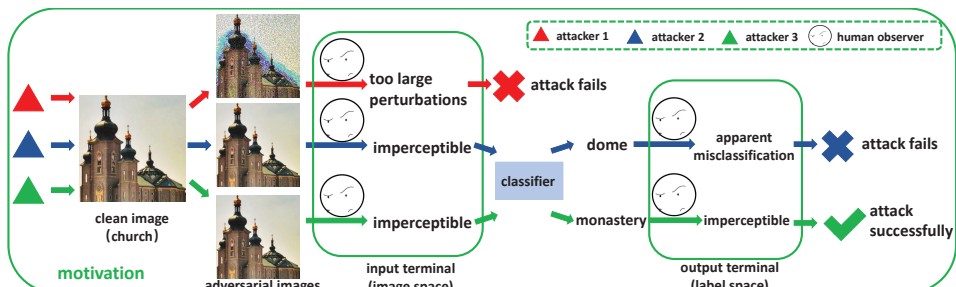

Figure 1: This graph illustrates the importance of the imperceptibility of adversarial samples in both image space and label space. Triangles are three attackers. Circles represent human observers.

The second requirement for the ideal attacker is the imperceptibility in the image space. This means the magnitude of perturbations in the pixel level needs to be as tiny as possible so that it is imperceptible to human eyes. For example, additive perturbations are minimized with $l_p$ norm to generate

imperceptible adversarial samples (Seyed-Mohsen et al., 2016). Extreme cases also exist where only changing one or a few pixels (Su et al., 2019; Modas et al., 2019) can make classifiers fail. Moosavi-Dezfooli et al. (2017) even show the existence of universal (image-agnostic) perturbations.

The third requirement for the ideal attacker, which is newly proposed in this paper, is the imperceptibility of the error made by the classifier in the label space. It means making the classifier to mis-classify an image as the label which is similar to its ground truth, so that people won't notice the misclassification. For example, in Figure 1, a human user will probably ignore the mis-classification if an attacker caused a "church" to be mis-classified as a "monastery" as the third attacker does. However, a human user will easily notice the mistake if an attacker caused a "church" to be mis-classified as a "dome" as the second attacker does or caused an apparent perturbation in the image space as the first attacker does. In real applications, a human user will take defensive measures as soon as he notices the attack. Therefore making the whole attack process imperceptible is crucial for letting observers' guard down. Tiny perturbations in the image space but large perturbations in the label space can muddle through on the input terminal. But as soon as observers check on the output terminal and see the obviously-incorrect label for an input, they will realize that the classifier fail due to some attacks and take defensive measures immediately, just as Figure 1 shows. This justifies the power of attacks which also confuse people in the label space. So the imperceptibility in the label space is quite important. However, to our best knowledge, few attackers have realized this point.

In this paper, we propose an untargeted-attack algorithm called LabelFool, to perturb an image to be mis-classified as the label which is similar to its ground truth, so that people won't notice the mis-classification. In the meantime, LabelFool also guarantees the imperceptibility in the image space as well as maintaining a high attack rate in fooling classifiers. There are two steps by which we accomplish our goal. The first step is to choose a target label which is similar to the input image's ground truth. The second step is to perturb the input to be classified as this target label. The way is finding the classification boundary between the current label and the target label, and then moving the input towards this boundary until it is classified as the target label. We conduct a subjective experiment on ImageNet (Deng et al., 2009) which shows that adversarial samples generated by our method are indeed much less recognizable in the label space by human observers than other attacks. We also perform objective experiments on ImageNet to demonstrate that adversarial samples generated by LabelFool still guarantee the imperceptibility in the image space as well as maintaining a high attack rate in fooling classifiers.

## 2 RELATED WORK

The phenomenon that neural networks are sensitive to adversarial samples was proposed by Szegedy et al. (2013). Since then, many researchers have studied how to generat adversarial samples. FGSM (Goodfellow et al., 2015) was proposed to maximize the classification error subject to $l_\infty$-norm based distortion constraints. CW attack (Carlini & Wagner, 2017) generates adversarial samples by solving an optimization problem based on $l_0/l_2/l_\infty$ constraint, and $l_0$ CW attack is the first proposed method that can cause targeted misclassification on the ImageNet dataset, meaning that we can specify the label of adversarial samples. But this designation of the target label is arbitrary. Until this paper, there has no guide about how to choose a target label such that it is difficult for a person to notice that the network has failed.

Besides achieving the goal of misclassification, many researchers realize the importance of imperceptibility in the image space (Xu et al., 2019). One-pixel attack (Su et al., 2019) and SparseFool (Modas et al., 2019) attack networks in a scenario where perturbing only one/a few pixels can make a big difference. Moosavi-Dezfooli et al. (2017) show the existence of universal image-agnostic perturbations for state-of-the-art deep neural networks. DeepFool (Seyed-Mohsen et al., 2016) seeks the minimum image-level distortion. And for generating adversarial samples, it directly moves the input sample to the nearest class in the feature space. This is the most closely related work to ours, because features extracted from classification models can reflect images' perceptual information and the classes which are close in the feature space are often perceptually similar. However, DeepFool approximates the multi-dimensional classification boundaries in two dimensions and this might make big errors on finding the nearest class. All these attacks generate adversarial samples by iteration and the algorithm stops as soon as an adversarial sample is born no matter what label it

belongs to. This will lead to an apparent misclassification so that observers will sound the defensive alarm quickly.

In this paper, we will compare our method with three attacks: FGSM, DeepFool and SparseFool to show the advantage of our method in the imperceptibility in the label space. We will also demonstrate that the performance gain in the label space is not at the expense of the loss in the image space or attack rate.

## 3   LABELFOOL

In this section, we will introduce our method about how to choose a target label which is undetectable by human observers and how we can perturb the input image so that the classifier assigns this specific label. The whole pipeline is shown in Figure 2. All the symbols and notations used in this paper are summarized in Table 1. We use the same notation $i(i = 1, 2, \dots)$ for "class" and "label", because "class" and "label" are interchangeable in this paper. LabelFool contains two steps. The first step is to choose a target label for the input image which is similar to its ground truth. The second step is to perturb the input image to be classified as this label. Inspired by DeepFool (Seyed-Mohsen et al., 2016), we make modifications at the feature level. We keep moving the input towards this chosen class at the feature level until it is classified as the label we want.

### 3.1   CHOOSE A TARGET LABEL

The first step of our method is choosing the target label $t_x$ for an input image $x$. As we want the target label to be imperceptible in the label space to human observers, we need to find the most "similar" label to the input image's ground truth $l_x$ where the most "similar" means the nearest in the perceptual distance metric. However, $l_x$ is usually unknown when an input image is given. So it is important to estimate the probability distribution $P$ of an input's ground truth $l_x$, based on which, we can compute the distance between each class in the dataset and $l_x$, then choose the nearest one as the target class. We propose a weighted distance model to achieve this goal. Before introducing the model, there are some preparations.

Given two image $x, y$, we choose pre-trained image classification models to extract features $\phi_x, \phi_y$ because these features can reflect some perceptual information. As we want to calculate the distance in the perceptual distance metric and cosine distance has been used to measure perceptual similarity in many works (Lin et al., 2016; Wang et al., 2019), we compute the distance between $x$ and $y$ as $d(x, y) = 1 - \cos(\phi_x, \phi_y)$. After having the distance between two images, we can compute the distance between classes. Each class is a set of images. To measure the distance between two sets, we choose Hausdorff distance (Henrikson, 1999). The distance between class $i$ and class $j$ is denoted as $D_{i,j}$. Suppose a dataset has $n$ classes. Then, we can construct a matrix $\boldsymbol{D} \in \mathbb{R}^{n \times n}$ by calculating the distance between all pairs of classes in the dataset, and it will be used in the following probability model to provide the distance we need. After these preparations, we can start to decide the target label for an input image.

As introduced before, we need to estimate the probability distribution $P$ of the ground truth $l_x$ because we want to find the nearest label to $l_x$ which is unknown in the beginning. When an image $x$ is put into a classifier $f$, state-of-the-art machine learning classifiers usually output a predicted label $\hat{l}_x$ and a probability vector $\hat{\boldsymbol{p}}$ whose elements mean $P(x \in class\ i) = \hat{p}_i$. For simplicity, we suppose the elements in $\hat{\boldsymbol{p}}$ are sorted in the descending order. Meanwhile, $\hat{\boldsymbol{p}}$ can be thought as $P$'s approximation. Furthermore, we define a distance function between $l_x$ and the class $i$ in a n-classes dataset as $D_i(l_x)$. In order to choose the nearest label to $l_x$ as the target label $t_x$, we need to estimate the expectation of $D_i(l_x)$ which is denoted as $\mathbb{E}_{l_x \sim P}[D_i(l_x)](i = 1, \dots, n)$. In general, our target function is Eq. (1).

$$t_x = \underset{i=1,\dots,n}{\arg\min} \mathbb{E}_{l_x \sim P}[D_i(l_x)] \tag{1}$$

Specifically, when $\hat{p}_1$ is larger than some threshold $\delta_1$, we use Maximum Likelihood Estimation (MLE) (Pfanzagl, 2011) which means we believe the classifier and take the predicted label $\hat{l}_x$ as

Table 1: Notations used in this paper.

| Symbol | Meaning |
|---|---|
| $x$ | Input image |
| $\phi_x$ | The feature of image $x$ |
| $\boldsymbol{D}$ | The perceptual distance matrix where $D_{i,j}$ represents the distance between class $i$ and class $j$ |
| $f$ | Classifier |
| $\hat{\boldsymbol{p}}$ | Probability vector (elements are in descending order) |
| $l_x$ | The ground truth of the input image $x$ |
| $\hat{l}_x$ | The predicted class of the input image $x$ by the classifier |
| $D_i(l_x)$ | A function calculating the distance between class $i$ and the ground truth $l_x$ |
| $t_x$ | Target label for input $x$ |
| $\delta_1, \delta_2$ | Two thresholds, in this paper, $\delta_1 = 0.8, \delta_2 = 0.01$ |
| $M$ | The number of elements larger than $\delta_2$ in $\hat{\boldsymbol{p}}$, i.e. $M = \max_{j=1,...,n}\{j : \hat{p}_j > \delta_2\}$ |
| $\mathcal{F}_j$ | The classification boundary between class $\hat{l}_x$ and class $j$ of an image |

the ground truth $l_x$, then choose the label (except $\hat{l}_x$) nearest to $\hat{l}_x$ as the target label $t_x$. So in this circumstance, we assume $l_x = \hat{l}_x$ and $\mathbb{E}_{l_x \sim P}[D_i(l_x)] = D_{i,\hat{l}_x}$. Therefore, $t_x$ is

$$t_x = \underset{i \neq \hat{l}_x, i=1,...,n}{\arg\min} D_{i,\hat{l}_x} \quad \text{if } \hat{p}_1 > \delta_1. \tag{2}$$

When $\hat{p}_1$ is smaller than the threshold $\delta_1$, we are not sure whether $l_x$ is equal to $\hat{l}_x$. Instead, we sample some labels and compute the weighted distance between each label $i$ and these labels. We sample all labels whose probability are larger than a threshold $\delta_2$ and we use $M$ to represent the number of sampled labels. We think the input image might belong to one of these $M$ labels. The labels whose probability are smaller than $\delta_2$ will be abandoned because we think the input image can hardly fall into these categories. The weight and the distance is provided by the vector $\hat{\boldsymbol{p}}$ and matrix $\boldsymbol{D}$ respectively. So in this circumstance, as we are not sure which label is the ground truth, we want to find a target label which has the minimum expected distance with all these possible labels. Therefore, the value of $\mathbb{E}_{l_x \sim P}[D_i(l_x)]$ can be approximated as $\sum_{j=1}^{M} \hat{p}_j \cdot D_{i,j}$ and the target label $t_x$ is shown in Eq. (3). This can be explained by **Importance Sampling** (Owen & Zhou, 2000) because it is hard to sample from the real probability distribution $P$. We can only use the probability distribution $\hat{\boldsymbol{p}}$ which is an approximation of $P$ to estimate the value of $\mathbb{E}_{l_x \sim P}[D_i(l_x)]$.

$$t_x = \underset{i=1,...,n}{\arg\min} \sum_{j=1}^{M} \hat{p}_j \cdot D_{i,j} \quad \text{if } \hat{p}_1 \leq \delta_1 \tag{3}$$

In conclusion, the whole strategy for choosing the target label $t_x$ of an input image $x$ is computed as Eq. (4). The target label $t_x$ minimizes $\mathbb{E}_{l_x \sim P}[D_i(l_x)]$ just as Figure 2 shows.

$$t_x = \begin{cases} \underset{i \neq \hat{l}_x, i=1,...,n}{\arg\min} D_{i,\hat{l}_x} & \text{if } \hat{p}_1 > \delta_1 \\ \underset{i=1,...,n}{\arg\min} \sum_{j=1}^{M} \hat{p}_j \cdot D_{i,j} & \text{otherwise} \end{cases} \tag{4}$$

## 3.2 GENERATE ADVERSARIAL SAMPLES

After having the target label, the second step is to attack the input image to be mis-classified as this target label. It's easy to achieve by taking the target label as a parameter and putting it into the targeted-attack algorithm such as targeted-FGSM (Goodfellow et al., 2015) and targeted-CW

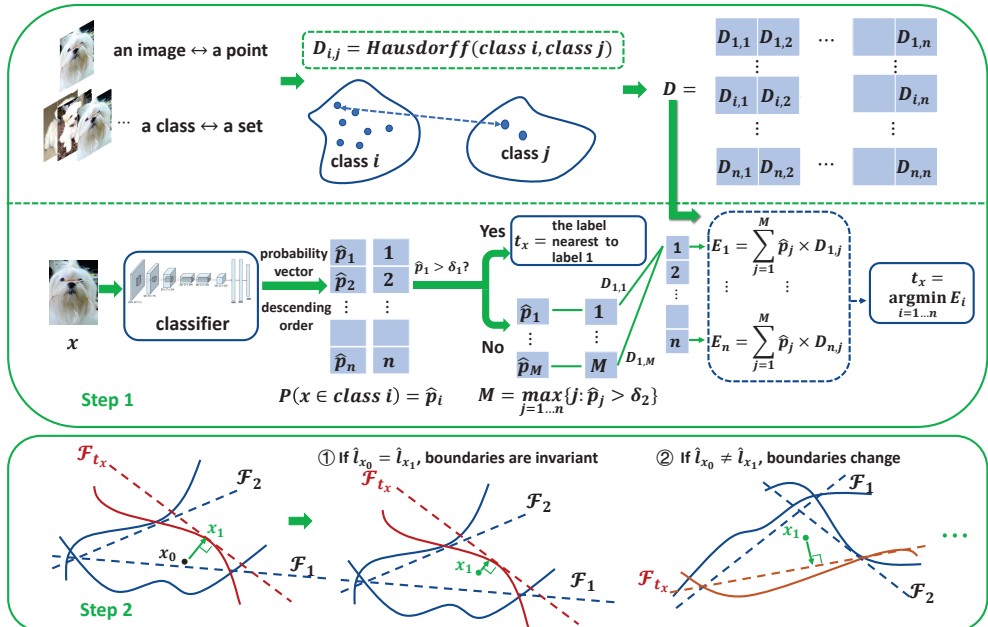

Figure 2: Pipeline of our method. There are two steps. **In the first step**, we first compute the distance $D_{i,j}$ between every two class $i, j$ in a n-classes dataset. Then we choose the target label $t_x$ for an input image $x$ by two strategies according to the value of $\hat{p}_1$. **The second step** is to attack the input into this target label. Solid lines are the real boundaries between the current label and the indicated label and dashed lines with notes $\mathcal{F}$ are the approximate two-dimensional boundaries. Red indicates the target label while blue indicates other labels. Our method moves the input towards the boundary $\mathcal{F}_{t_x}$ until it is classified as $t_x$.

(Carlini & Wagner, 2017), but this operation may suffer huge loss in the image space because of large perturbations.

Inspired by DeepFool (Seyed-Mohsen et al., 2016), we propose a method which can not only attack an input image to be mis-classified as the target label successfully, but also ensure tiny perturbations in the image space. The mathematical derivation in this step is similar to DeepFool (Seyed-Mohsen et al., 2016) and the only difference is that, we have a target label chosen in the first step while DeepFool doesn't. As introduced in DeepFool (Seyed-Mohsen et al., 2016), a high dimensional classification boundary can be approximated by a line in two dimensions. As shown in Figure 2, for an image $x_0$, $\mathcal{F}_j$ represents the 2D approximated boundary between its current predicted class and class $j$ and $t_x$ is the target class we choose in the first step. In the first iteration, we move $x_0$ towards $\mathcal{F}_{t_x}$ and get a new point $x_1$. The direction of movement is perpendicular to $\mathcal{F}_{t_x}$. The distance of the movement is the vertical distance from $x_0$ to $\mathcal{F}_{t_x}$. If the predicted label $\hat{l}_{x_1}$ of the new point equals $\hat{l}_{x_0}$, the classification boundaries are the same as those before moving $x_0$. Otherwise, the classification boundaries change. No matter whether the boundaries change or not, we repeatedly move the current point towards $\mathcal{F}_{t_x}$ until it is classified as label $t_x$ or the maximum number of iterations has been reached. A pseudocode of the second step is shown in Algorithm 1.

---

**Algorithm 1** : Generate Adversarial Samples

**Input:** image $x$, classifier $f$, target label $t_x$
**Output:** Adversarial image $\hat{x}$
1: initialize $x_0 \leftarrow x, i \leftarrow 0$
2: **while** $\hat{l}_{x_i} \neq t_x$ and $i < max\_iter$ **do**
3:     $w \leftarrow \nabla f_{\hat{l}_{x_i}}(x_i) - \nabla f_{t_x}(x_i)$
4:     $g \leftarrow f_{\hat{l}_{x_i}}(x_i) - f_{t_x}(x_i)$

5:     $r_i \leftarrow \dfrac{|g|}{\|w\|_2^2} w$
6:     $x_{i+1} \leftarrow x_i + r_i, i \leftarrow i+1$
7: **end while**
8: **return** $\hat{x} = x_{i+1}$

---

## 4 EXPERIMENTS

In this paper, all experiments are conducted on ImageNet. ImageNet provides the CLS-LOC dataset for classification tasks. Its train split contains about 1300 thousand images. There are 50 thousand validation images and 100 thousand test images. Our experiments are conducted on the train split of CLS-LOC dataset which will be noted as ImageNet-train split in the following part. We perform extensive experiments to show LabelFool can satisfy all three levels of requirements as an attacker. First we demonstrate the deceptiveness of samples generated by LabelFool to humans in the label space through a subjective experiment. Then we calculate the perceptibility and image quality of adversarial samples to show there is not much loss in the image space even compared to DeepFool (Seyed-Mohsen et al., 2016), which is the state-of-the-art method in the image space. Finally, we conduct attacks on several models to prove the exceptional ability of our method on fooling neural networks which is the first requirement for an ideal attack.

### 4.1 IMPERCEPTIBILITY IN THE LABEL SPACE

**Setup.** In this part, we will compare LabelFool with three attack methods: DeepFool, FGSM and SparseFool. We first sample 600 source images from ImageNet-train split randomly, and these images are in different classes. Each source image will derive four adversarial images and a baseline image, namely DeepFool-attacked image, LabelFool-attacked image, FGSM-attacked image, SparseFool-attacked image and clean image. Each adversarial image has its mis-classified label and each baseline image has the truth label. We then use term "puzzles" to describe the combination of an image and its label, for $5 \times 600 = 3000$ puzzles. A human observer needs to determine whether the label is correct for the image, answering "True" or "False" for each puzzle. To eliminate observers' memory effects, we split 3000 puzzles into five groups, ensuring that 600 images in one group come from different source images. An interface presentation of our subjective experiment is shown in Appendix A. We have 10 observers (3 females and 7 males, age between 20-29) to do our subjective experiment.

**Evaluation.** In this paper, we define an index named *Performance Gain (PG)* as the evaluation index. Human observers answer "True" or "False" for each puzzle, and the rate with which they answer incorrectly is called *Confusion Rate (CR)*. So every observer has a *CR* for each attack method or baseline. It is an absolute indicator demonstrating how much observers are confused by a set of puzzles. But doing arithmetic on *CR* of different observers is meaningless, as different observers have different baseline results. So we define a relative indicator called *Performance Gain (PG)*, which demonstrates how much improvement in the confusion rate after attacking comparing with baseline. It is a kind of normalization. The formula for PG is shown in Eq. (5), where $CR_A$ means the confusion rate of an attacker and $CR_B$ means the confusion rate of baseline. The higher *PG* an attacker has, the better it confuses people in the label space.

$$PG_A = \frac{CR_A - CR_B}{CR_B} \ (A \ is \ an \ attacker, \ B = Baseline) \tag{5}$$

**Results.** We report the average performance gain of 10 observers in total images in the left of Figure 3. As a whole, there is a huge improvement compared with FGSM and SparseFool, about 25 percent improvement and 30 percent improvement respectively. Compared with DeepFool, the gap in *PG* is a little smaller because DeepFool finds the nearest class in the feature level and features usually reflect images' perceptual information as we introduced in Section 2. But there is still 3 percent improvement in performance gain.

As animal classes are more fine-grained, the effects of the imperceptibility in the label space become more pronounced. We attack an animal image so that its true animal label changed into a similar animal label, it is difficult for humans to notice that our attack is taking place. Meanwhile, other attacks change the label into an obviously-incorrect label such as a non-animal category or another species (Some examples in Appendix B). In our subjective experiment, there are 247 animal images out of 600 source images. The right graph in Figure 3 shows the average performance gain of 10 observers in animal images. The accurate data for both graphs in Figure 3 is shown in Appendix C.

As for animal images, the improvement is very obvious comparing with all three attack methods. There are about nearly 90 percent improvement in performance gain comparing with FGSM and

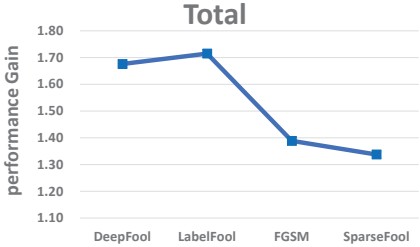 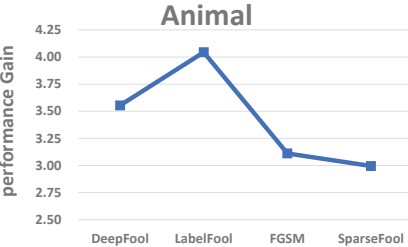

Figure 3: A line chart for average performance gain of 10 observers. The horizontal axis represents four attack methods. The vertical axis represents the mean value of 10 human observers' performance gain. The graph in the left is for total results, and the right one is for animal images.

SparseFool. A significant improvement can also be seen when comparing with DeepFool, there are about 50 percent improvement in average performance gain.

## 4.2 IMPERCEPTIBILITY IN THE IMAGE SPACE

In this subsection, we will show our performance in the image space to demonstrate that our improvement in the label space is not at the cost of huge loss in the image space. We use three metrics to evaluate performance in the image space. One is perceptibility which is similar to the definition in previous works (Szegedy et al., 2013; Seyed-Mohsen et al., 2016) $:p = \frac{1}{W_N \times H_N} \sum_{w=1}^{W_N} \sum_{h=1}^{H_N} \|\Delta y_{w,h}\|^2$, where $y_{w,h}$ is a 3-dimensional vector representing the RGB intensities (normalized in [0, 1]) of a pixel. The other two are perceptual similarity (Zhang et al., 2018) and PieAPP (Prashnani et al., 2018). These two are metrics for image quality. Perceptual similarity measures the perceptual distance between an image and its reference image while PieAPP measures the perceptual error. In this paper, the reference image is the clean image. And the smaller these three metrics are, the better the adversarial samples are.

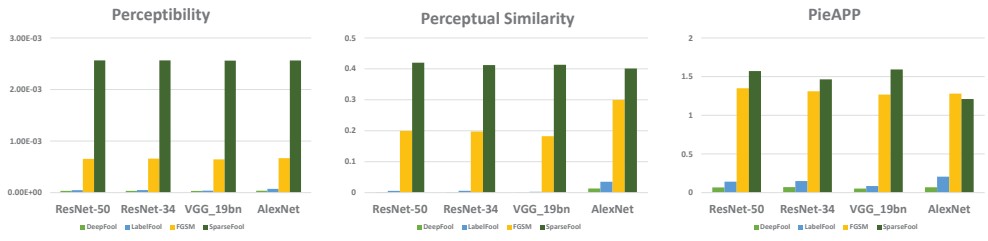

Figure 4: Mean value of perceptibility, perceptual similarity and PieAPP for adversarial samples generated by different attack methods on different models.

We randomly choose 1000 images from ImageNet and attack the classifier to generate 1000 adversarial samples. Then we compute mean value for these adversarial samples of three metrics. In this experiment, we test four classifiers: ResNet-34, ResNet-50, VGG-19 (with batch normalization) (Simonyan & Zisserman, 2014) and AlexNet (Krizhevsky et al., 2012a). The results are shown in Figure 4 whose original data are reported in Appendix C. We can see although LabelFool is significantly better than FGSM and SparseFool, it is still a little worse than DeepFool in all three metrics. However, visual results (Figure 5) indicate that human observers can not notice the difference between LabelFool and DeepFool in the image space as the metric value is on such a small scale.

## 4.3 FOOL NETWORKS

We will show attack rate in the last experiment which is the most fundamental requirement for an attacker. Results are shown in Table 2. The results are the average value of three groups of

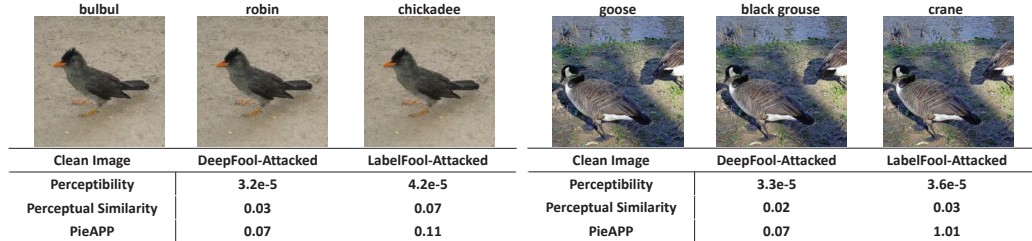

| Clean Image | DeepFool-Attacked | LabelFool-Attacked | | Clean Image | DeepFool-Attacked | LabelFool-Attacked |
|---|---|---|---|---|---|---|
| Perceptibility | 3.2e-5 | 4.2e-5 | | Perceptibility | 3.3e-5 | 3.6e-5 |
| Perceptual Similarity | 0.03 | 0.07 | | Perceptual Similarity | 0.02 | 0.03 |
| PieAPP | 0.07 | 0.11 | | PieAPP | 0.07 | 1.01 |

Figure 5: Two visual results of adversarial samples generated from AlexNet in the image space. In each result, the left image is the clean image, the middle one is DeepFool-attacked adversarial sample and the right one is LabelFool-attacked adversarial sample. Above the images are the true label/ the label after attacked. The three metrics of adversarial samples are reported in the table below them.

Table 2: Attack rate of different methods on different models.

| Model | DeepFool | LabelFool | FGSM | SparseFool |
|---|---|---|---|---|
| ResNet-34 | 92.67% | **97.50%** | 95.03% | 92.60% |
| ResNet-50 | 93.08% | **97.88%** | 95.09% | 92.53% |
| VGG-19(bn) | 92.03% | **97.48%** | 94.59% | 83.70% |
| AlexNet | 90.35% | **97.38%** | 96.44% | 89.11% |

experiments. Each group has 1000 original images from ImageNet, we use these original images to generate adversarial images for four models respectively. We surprisingly find that LabelFool has the highest attack rate on all models comparing with other methods. This might benefit from our probability model which is used to choose the target label. Because in our strategy, when $\hat{p}_1 \leq \delta_1$, we do not use the predicted label as the ground truth like other methods do. Instead, we consider all labels whose probability are larger than $\delta_2$ and choose the label nearest to all these labels as the target label. This operation can avoid some mistakes and improve the attack rate when the classifier doesn't give a correct classification result. An example is shown in Appendix D.

## 5 CONCLUSION AND FURTHER DISCUSSION

**Conclusion.** In this study, we pay attention to tiny perturbations in the label space. To our best knowledge, we are the first one who points out the importance of the imperceptibility in the label space for adversarial samples. Furthermore, we explore a feasible method named LabelFool to identify a target label "similar" with an input image's ground truth and perturb the input image to be mis-classified as this target label so that a human observer will overlook the misclassification and lower the vigilance of defenses. Our experiments show that, while LabelFool is a little behind DeepFool in the image space, it is much imperceptible in the label space to human observers. Since we adopt Importance Sampling instead of MLE only in traditional method, the success rate of attack also get gains.

**Further discussion.** In this paper, we just propose a feasible way to generate adversarial samples which can confuse people in the label space. However, there is room for improvement in our approach. Our results provide the following avenues for future research.

- The perceptual features can be optimized by a well-designed loss function which can improve the accuracy rate in finding nearest label ulteriorly.

- We only consider perceptual distance in this paper, but semantic distance also has its significance for reference of confusing people in the label space. We may take the semantic tree into consideration and make a trade off between perceptual distance and semantic distance in future research.

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

## A   AN INTERFACE PRESENTATION OF THE SUBJECTIVE EXPERIMENT

Figure 6 shows the interface of our subjective experiments.

Figure 6: Interface presentation of our subjective experiment.

## B   EXAMPLES FOR ANIMAL CLASSES

Figure 7 shows three examples for animal classes to demonstrate that LabelFool makes fine-grained changes but other methods make some ridiculous changes instead.

| Image | Ground truth | DeepFool | LabelFool | FGSM | SparseFool |
|---|---|---|---|---|---|
|  | cock | partridge | hen | partridge | partridge |
|  | peacock | sea urchin | gallinule | wool | poncho |
|  | fire salamander | sea slug | spotted salamander | coral reef | courgette |

Figure 7: Three examples for animal images. The first column shows the clean image. The second column shows the ground truth label and other columns show the label after attacked.

## C  ORIGINAL DATA FOR FIGURE 3 AND 4

Table 3 is the original data for Figure 3. The original data of perceptibility, perceptual similarity, PieAPP in Figure 4 is reported in Table 4, 5, 6 respectively. It is provided for the sake of convince if anyone wants to rewrite Figure 3 or 4.

Table 3: Data for Figure 3

|  | Index | DeepFool | LabelFool | FGSM | SparseFool |
|---|---|---|---|---|---|
| **Total** | Performance Gain | 1.68 | **1.71** | 1.39 | 1.34 |
| **Animal** | Performance Gain | 3.55 | **4.04** | 3.11 | 3.00 |

Table 4: Perceptibility Data for Figure 4

| Model | DeepFool | LabelFool | FGSM | SparseFool |
|---|---|---|---|---|
| ResNet-50 | 3.36E-05 | 4.75E-05 | 6.55E-04 | 2.56E-03 |
| ResNet-34 | 3.35E-05 | 4.84E-05 | 6.59E-04 | 2.56E-03 |
| VGG-19bn | 3.18E-05 | 3.80E-05 | 6.44E-04 | 2.56E-03 |
| AlexNet | 3.59E-05 | 7.24E-05 | 6.68E-04 | 2.56E-03 |

Table 5: Perceptual Similarity Data for Figure 4

| Model | DeepFool | LabelFool | FGSM | SparseFool |
|---|---|---|---|---|
| ResNet-50 | 1.48E-3 | 5.64E-3 | 0.20 | 0.42 |
| ResNet-34 | 1.39E-3 | 5.91E-3 | 0.20 | 0.41 |
| VGG-19bn | 6.86E-4 | 2.54E-3 | 0.18 | 0.41 |
| AlexNet | 1.34E-2 | 3.50E-2 | 0.30 | 0.40 |

Table 6: PieAPP Data for Figure 4

| Model | DeepFool | LabelFool | FGSM | SparseFool |
|---|---|---|---|---|
| ResNet-50 | 0.07 | 0.14 | 1.35 | 1.57 |
| ResNet-34 | 0.07 | 0.15 | 1.31 | 1.46 |
| VGG-19bn | 0.05 | 0.08 | 1.27 | 1.59 |
| AlexNet | 0.07 | 0.21 | 1.28 | 1.21 |

## D  AN EXAMPLE FOR SECTION 4.3

This is an example to illustrate why our method has the highest attack rate. We only give an example of DeepFool and LabelFool. SparseFool and FGSM have similar effects with DeepFool.

Figure 8 is an example where the classifier fail to give a correct classification for the input image $x$. The ground truth of $x$ is class 2 while the predicted class is class 3. In this example, DeepFool takes class 3 as the true class. Then DeepFool finds the nearest class to class 3 in the feature space which is class 2 in this example, and moves the input image towards class 2. When the perturbed image is classified as class 2 which is different from the predicted class, DeepFool considers the attack

succeed and stops the algorithm. However, it fails to attack actually because class 2 is the true class of $x$.

Different from DeepFool, LabelFool sample top 3 classes in this example because their probabilities are larger than 0.01 and compute the expected distance between each class in the dataset and these 3 classes (Eq.(3)). Finally, LabelFool choose class 394 as the target class because it has the minimum expected distance with top 3 classes. By moving $x$ towards class 394, LabelFool attacks successfully.

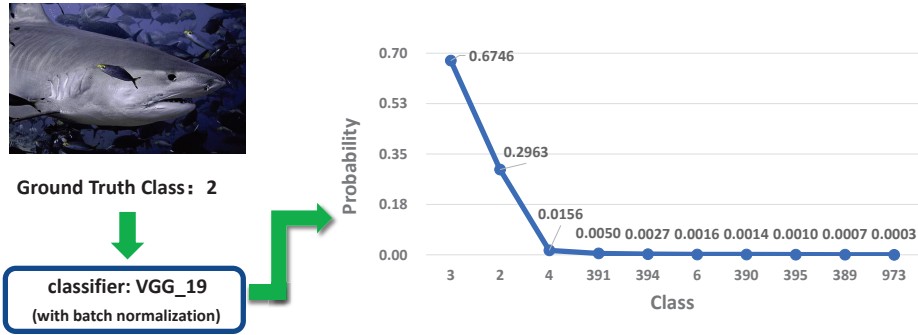

Figure 8: The line chart in the right part shows the top 10 elements in $\hat{p}$, when an image $x$ whose ground truth class is class 2 is given into the classifier VGG_19bn. The horizontal axis represents the class and the vertical axis represents the probability that $x$ belongs to this class. The predicted class $\hat{l}_x$ is class 3 and it means the classifier fail to give a correct classification.

## E    SUPPLEMENTARY EXAMPLES FOR LABELFOOL

Figure 9 shows some supplementary examples to show what LabelFool actually does.

| Image | Ground truth | DeepFool | LabelFool | FGSM | SparseFool |
|---|---|---|---|---|---|
| | tench | sea cucumber | coho | barracouta | barracouta |
| | whiptail lizard | banded gecko | alligator lizard | stole | damselfly |
| | harvestman | wolf spider | long-horned beetle | wolf spider | wolf spider |
| | beer bottle | pop bottle | pop bottle | pop bottle | wine bottle |
| | chain saw | lumbermill | power drill | lumbermill | lumbermill |
| | liner | dock | container ship | dock | dock |

Figure 9: Some other examples for illustrating what LabelFool does. The first column shows the clean image. The second column shows the ground truth label and other columns show the label after attacked.

# F AN APPLICATION: FACE RECOGNITION

Figure 10 is an application to show it is necessary to generate imperceptible adversarial examples in the label space even for image classification tasks. We take face recognition system for entrance as an example. In Figure 10, A is the person who is using the face system to go into the gate. LabelFool aims to let the system misclassify A and B who is the one looks like A, but other untargeted attacks may let the system misclassify A an C who looks totally different from A. The attack is easy to be detected by the guard if the system misclassifies A and C, but it is hard to detect if the system misclassifies A and B. Letting a fake B in will bring great potential risks to security and safety.

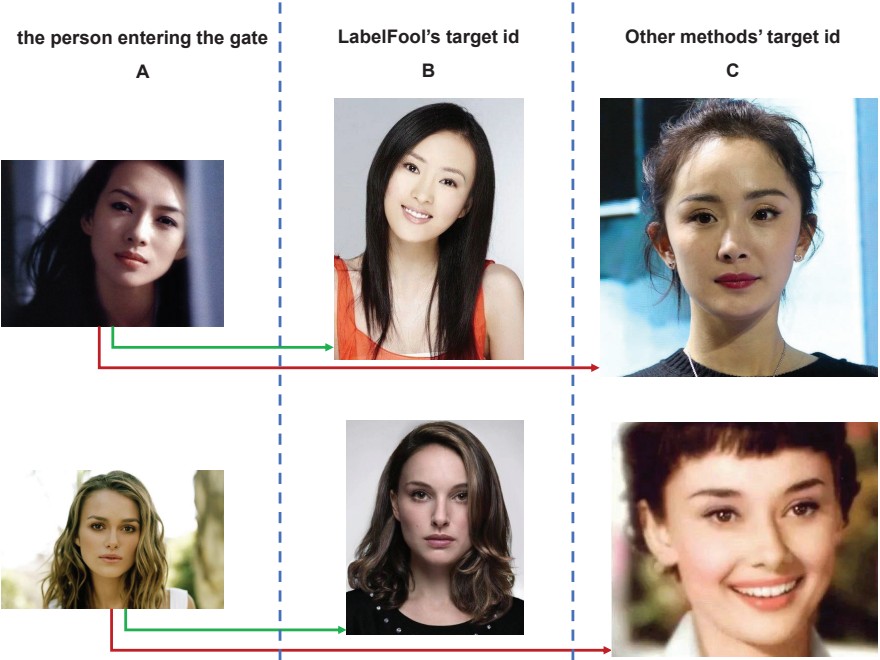

Figure 10: A is the person who is using a face system to enter the gate. Green lines represent what LabelFool aims to do, that is to miscalssify A and B who looks like A. Red lines represent what other untargeted attacks do. They misclassify A and C who looks totally different from A and this error is easy to be detected by the guard.

