# OpenReview forum: "LabelFool: A Trick in the Label Space"
_ICLR.cc/2020/Conference — Reject_

### Official Review · AnonReviewer3 · 2019-10-21
**Official Blind Review #3**

**Rating:** 1

**Review:**

This paper proposes a method for constructing adversarial attacks that are less detectable by humans, by changing the target class to be a class similar to the original class of the image. The resulting attack methodology is then studied in terms of its imperceptibility in label space, and shown to be less perceptible in label space to human observers, while not coming at a cost in image space.

The paper presents compelling evaluation of the method and does seem to succeed in proving that their proposed attack satisfies the stated goal. However, it appears as though this goal is somewhat counter to the main point of adversarial examples---indeed, if the label is reasonable to a human, then what makes the adversarial example adversarial? The main threat in adversarial examples research seems to be that it is possible to induce predictions that are arbitrarily different from humans' on natural-looking in puts. Thus, changing the label to something that a human actually agrees with would actually reduce the impact of the adversarial attack.

In order to improve the paper, I would suggest applying the same (or similar) methodologies to other areas of ML security where imperceptibility in label space is commonly desired---for example, in data poisoning attacks or backdoor attacks. In general, such attacks are much more likely to be "inspected" by humans, and so imperceptibility in both label and image space is very desirable. However, I suspect that this would require significant effort and changes to the paper, and so for now I recommend rejection.

**Experience Assessment:**

I have published in this field for several years.

**Review Assessment: Checking Correctness Of Derivations And Theory:**

N/A

**Review Assessment: Checking Correctness Of Experiments:**

I carefully checked the experiments.

**Review Assessment: Thoroughness In Paper Reading:**

I read the paper thoroughly.

---

> ### Author Response · Authors · 2019-11-08
> **Response to Reviewer #3**
>
> We thank the reviewer for the time and some good suggestions, and would like to answer the reviewer’s questions as follows:
>
> First, there is some misunderstanding about “adversarial examples”. This concept was first proposed in 2014 [1]. It is defined as “We find that applying an imperceptible non-random perturbation to a test image, it is possible to arbitrarily Change the Network’s Prediction. … We term the so perturbed examples ‘adversarial examples’.” in work [1]. Many other works [2,3] follow this definition that adversarial examples are the ones misclassified by the network （by adding imperceptible perturbations）. Meanwhile, to the best of our knowledge, all works in adversarial attack community report the “success rate of attack” as the misclassification rate of their target model without checking whether the generated examples are reasonable to humans or not. As long as the prediction is different from the ground truth, the perturbed example is “adversarial” no matter it is reasonable to human or not.
>
> Second, we think there are two evaluations of “the impact of the adversarial attack”. One is whether the error happens, it is evaluated by the success rate of attack (misclassification rate of the target model). The other is how long the error lasts, it is evaluated by the time that a human user detects the attack. LabelFool does as well as other attacks on the first evaluation, but does better on the second evaluation because it is less detectable by a human user.
>
> Third, we only conduct experiments on the image classification task and you think this task is less likely to be “inspected” by humans. But the fact is, this task is widely used in many fields in our life. Please see Figure 9 in Appendix E in the new version. Taking face recognition system for entrance as an example, there will usually be a guard as “a human inspector” to check whether the man A who is entering the gate is the same as the system shows. In this case, if a model just misclassifies A and C, the person who looks totally different from A, the attack will be detected easily. But LabelFool aims to let the model misclassify A and B who looks like A as shown in Figure 9. If the guard doesn’t identify carefully, he will let a fake B in and this error brings great potential security risks. So as an attacker, it is necessary to generate imperceptible examples in the label space for this task.
>
> Fourth, we appreciate your suggestion about applying this method to other areas of ML security, we will do this in our future work, but now, we just introduce this idea, provide a tool and conduct experiments on simple tasks to prove the feasibility of this idea.
>
> If we misunderstood you in this rebuttal or you still have questions about our motivation, please don’t hesitate to let us know. We are here to answer your questions all the time.
>
> References:
> [1] Szegedy et al. Intriguing properties of neural networks. (ICLR 2014)
> [2] Kurakin et al. Adversarial examples in the physical world[J]. arXiv preprint arXiv:1607.02533, 2016.
> [3] Goodfellow et al. Explaining and harnessing adversarial examples. (ICLR 2015)

---

> > ### Comment · AnonReviewer3 · 2019-11-13
> > **Response**
> >
> > I appreciate the authors clarification, but in the demonstrated settings, I can't imagine a case where perturbing an image of a "church" to be classified as a "monastery" is an adversarial example that one would care to generate.
> >
> > I'm also confused by how this approach meaningfully differs from just a targeted attack towards the second-highest label?

---

> > > ### Author Response · Authors · 2019-11-13
> > > **Response to Reviewer #3**
> > >
> > > We thank the reviewer for the comments, and would like to answer the reviewer’s questions as follows:
> > >
> > > First, as last comment said, please read the example in Figure 9 in Appendix E in our new version. And we have illustrated a real life case where LabelFool is needed in Appendix E. We just give an example in Figure 1 to help the readers understand what LabelFool actually does in theory. We can change the examples in Figure 1 into the examples in Figure 9 if you think examples in Figure 9 are more helpful for readers to understand our motivation.
> > >
> > > Second, we have considered about using the second-highest label as the target label, but we found some drawbacks in doing so. On the one hand, when the network’s confidence score for the highest class is higher than $\delta_1$ (we use 0.8 in this paper), there is no significant difference in confidence scores for other classes, so choosing the second-highest label is not appropriate. On the other hand, when the network’s confidence score for the highest class is lower than $\delta_1$, the second-highest label is usually the ground truth label (In Figure 8 in Appendix D). It will reduce the attack rate if we choose the second-highest label as the target label. So choosing the second-highest label is not appropriate in this case either. Therefore, we don’t use the second-highest label as the target label, but design an explicit method instead.
> > >
> > > If you still have questions about our motivation, please don’t hesitate to let us know. We are here to answer your questions all the time.

---

> > > > ### Comment · AnonReviewer3 · 2019-11-14
> > > > **Response**
> > > >
> > > > The motivation in the Appendix does make things slightly more clear, although to me still quite contrived.
> > > >
> > > > Still, even if one was to accept the motivation, I think without explicitly comparing to a naive strategy where one just does a targeted attack towards the second label (or, more accurately, to the highest-confidence incorrect label), the study is critically incomplete. (I had initially misunderstood the paper as using this as a baseline, which is why I did not include it in the initial review.)

---

> > > > > ### Author Response · Authors · 2019-11-14
> > > > > **Response to Reviewer #3**
> > > > >
> > > > > Let’s clarify our motivation again. We think it is important for attackers to design less detectable adversarial examples. Are there any attackers who would like their attack to be easily detected by users? And if adversarial attacks are less likely to be “inspected” by humans, why many previous works generate adversarial examples “imperceptible to human eyes” in the image space? Therefore, we think it is common and natural for attackers to have this requirement where they would like to generate less detectable adversarial examples. In our opinion, subtle mistakes can last for a long time so that the attack can bring huge potential risks a human can’t imagine, while obvious mistakes will be corrected in time so that the attack doesn’t have time to cause damage to the user.

---

### Official Review · AnonReviewer2 · 2019-10-22
**Official Blind Review #2**

**Rating:** 3

**Review:**

This paper describes a technique for creating adversarial images where the added perturbations are not only imperceptible to machines, but also to human observers. The authors describe why this might be beneficial. The method works by finding labels that are not too far from the source image's ground-truth labels, and moving the source image in that direction. To find the target label, the authors use a threshold on the confidence of predicted ground-truth labels. The authors test their algorithm using a newly proposed metric of how much a method allows imperceptibility for a human observer. They show that their method creates images whose perturbations are more impercetible to humans, compared to other methods, but are also imperceptible to machines.

My concern is as follows: If the misclassification is between A and B, and they are related classes, is the attack so bad? And what are the scenarios in practice when a user simply wants to create an attack, without regards to the target label chosen? I imagine normally the attacker has a target label in mind, so the part of the paper that chooses a target label is not very useful; and this is the main element of novelty, since the rest of the method is from DeepFool, as the authors explain. Some specific use cases of this methods should be discussed.

Minor suggestion: It would be useful to see examples like in Fig. 5 but with the classes (true/target) listed.

**Experience Assessment:**

I do not know much about this area.

**Review Assessment: Checking Correctness Of Derivations And Theory:**

I assessed the sensibility of the derivations and theory.

**Review Assessment: Checking Correctness Of Experiments:**

I assessed the sensibility of the experiments.

**Review Assessment: Thoroughness In Paper Reading:**

I read the paper thoroughly.

---

> ### Author Response · Authors · 2019-11-08
> **Response to Reviewer #2**
>
> We thank the reviewer for the time and concern, and would like to answer the reviewer’s questions as follows:
>
> First, let’s clarify the concept of adversarial examples. The concept of “adversarial examples” was first proposed in 2014 [1] and defined as “We find that applying an imperceptible non-random perturbation to a test image, it is possible to arbitrarily Change the Network’s Prediction. … We term the so perturbed examples ‘adversarial examples’.” That is to say, adversarial examples are the ones misclassified by the network (by adding imperceptible perturbations). As long as the prediction is changed by the perturbation, the perturbed example is called “adversarial example” no matter how the prediction is far from the ground truth. Therefore, we think that each adversarial example should be treated equally. There is no “good” or “bad” attack. There is only “successful” and “failed” attack.
>
> Second, to the best of our knowledge, most papers in adversarial attack community focus on untargeted attacks. Unfortunately, exiting untargeted attacks just focus on getting the networks failed (with imperceptible perturbations in the image space). They don’t care about the misclassified-label in the label space so that the misclassified-label may be very unreasonable. However, many effective defense methods [2,3] have been published. Unreasonable labels will cause the users detect the attack and take defensive measures to let the attack fail (just as introduced in Section 1). So “creating an attack without regards to the target label” is adverse for an attacker. Therefore, our method is applicable when an attacker conducts untargeted attack and hope it not to be detected. We recommend to create carefully automatically designed target label when doing untargeted attacks.
>
> Third, in Figure 7 in Appendix B of the original version, there are some figures as you suggested. In Appendix E of our new version, we add more examples in Figure 9. We also add true/target labels in Figure 5 as you suggested.
>
> If you still have questions about our motivation, please don’t hesitate to let us know. We are here to answer your questions all the time.
>
> References:
> [1] Szegedy et al. Intriguing properties of neural networks. (ICLR 2014)
> [2] Jia X, et al. ComDefend: An Efficient Image Compression Model to Defend Adversarial Examples. (CVPR 2019)
> [3] Dubey, Abhimanyu, et al. Defense against adversarial images using web-scale nearest-neighbor search. (CVPR 2019)

---

### Official Review · AnonReviewer4 · 2019-11-06
**Official Blind Review #4**

**Rating:** 3

**Review:**

This paper proposes a method to create adversarial perturbations whose target labels are similar to their ground truth. The target labels are selected using an existing perceptual similarity measure for images.  Perturbations are generated using a DeepFool-like algorithm. Human evaluation supports that the pair of the generated images and target labels are more natural to humans than prior attack algorithms.

This paper should be rejected due to the lack of motivation to create adversarial examples less detectable by humans automatically. Attackers can manually select target labels and apply targeted attacks. In the target label selection, attackers can choose less detectable labels if necessary. It is encouraged to provide some applications where attackers want to create less detectable adversarial examples in label space without manually assigning target labels.

==========
Update:

After reading the authors' responses, the motivation of the paper became clearer. I will not get surprised if this paper is accepted. However, all reviewers still share concerns about the importance of the problem tackled. I think the paper needs to suggest more applications and emphasize the value of the goal in the main paper before being published.

**Experience Assessment:**

I have published one or two papers in this area.

**Review Assessment: Checking Correctness Of Derivations And Theory:**

I assessed the sensibility of the derivations and theory.

**Review Assessment: Checking Correctness Of Experiments:**

I assessed the sensibility of the experiments.

**Review Assessment: Thoroughness In Paper Reading:**

I made a quick assessment of this paper.

---

> ### Author Response · Authors · 2019-11-08
> **Response to Reviewer #4**
>
> We thank the reviewer for the time , and would like to answer the reviewer’s questions as follows:
>
> First, we want to explain our motivation for this work. Our method is an untargeted attack, not a targeted attack. To the best of our knowledge, most papers in adversarial attack community focus on untargeted attacks. But exiting untargeted attacks don’t care about the misclassified-label in the label space so that the misclassified-label may be very unreasonable. However, many effective defense methods [1,2] have been published. Unreasonable labels will cause the users detect the attack and take defensive measures to let the attack fail (just as introduced in Section 1). So creating an attack without regards to the target label is adverse for an attacker. Therefore, we propose this LabelFool method in order to help an untargeted attack not to be defensed. We recommend to create carefully automatically designed target label when doing untargeted attacks
>
> Second, there are many applications where “attackers want to create less detectable adversarial examples in label space without manually assigning target labels”. Please see Figure 9 in Appendix E in the new version. Taking face recognition system for entrance as an example, there will usually be a guard as “a human inspector” to check whether the man A who is entering the gate is the same as the system shows. In this case, if the model just misclassifies A and C, the person who looks totally different from A, the attack will be detected easily. But LabelFool aims to let the model misclassify A and B who looks like A as shown in Figure 9. If the guard doesn’t identify carefully, he will let a fake B in and this error brings great potential security risks. So as an attacker, it is necessary to generate imperceptible examples in the label space for this task. But how does the attacker know who is the one looks like A in such a huge face database? LabelFool can help him much in this application.
>
> Third, we appreciate your suggestion about applying this method to other applications, we will do this in our future work, but now, we just introduce this idea, provide a tool and conduct experiments on simple tasks to prove the feasibility of this idea.
>
> If you still have questions about our motivation, please don’t hesitate to let us know. We are here to answer your questions all the time.
>
> References:
> [1] Jia X, et al. ComDefend: An Efficient Image Compression Model to Defend Adversarial Examples. (CVPR 2019)
> [2] Dubey, Abhimanyu, et al. Defense against adversarial images using web-scale nearest-neighbor search. (CVPR 2019)

---

> > ### Comment · AnonReviewer4 · 2019-11-13
> > **Response**
> >
> > Thank you for your response. The authors' explanation of LabelFool's application did make the motivation easier to understand. It increased my score to borderline, but it is still negative.
> >
> > Comments:
> > 1) (Comment to authors' response 1) I do not think that LabelFool should be categorized as an untargeted-attack. It will be more appropriate to classify it as a target-label selection algorithm for targeted-attack algorithms in (restricted) untargeted-attack settings. This categorization issue is not critical for my score.
> > 2) I think the key contribution of this paper is introducing the necessity of automatic and less-detectable target label selection algorithms. Thus, for my assessment, it is critical how much the problem will be important and how well the paper will motivate the machine learning community to discuss on the topic. The authors' response addressed to this point to some extent. Hence I slightly adjusted my score, but it was not so significant to flip my score into weak accept. Proposing more applications or performing evaluations in the settings of the proposed applications might have improved my score. I think these changes require significant modification of the paper, and I suggest for rejection this time.
> > 3) In terms of the significance of the proposed algorithm, I did not think it is beyond the bar of top-venues such as ICLR. It would have increased the significance if there were additional experiments that show that the label selection algorithm and targeted attack algorithm used in LabelFool performs better than other possible candidates. I am sorry that I could not provide this suggestion in the initial review.

---

> > > ### Author Response · Authors · 2019-11-14
> > > **Response to Reviewer #4**
> > >
> > > We thank the reviewer for increasing the score to borderline, but we could not see the rating changed in the system. We would like to answer the reviewer’s questions as follows:
> > >
> > > We are a little confused about “show that the label selection algorithm and targeted attack algorithm used in LabelFool performs better than other possible candidates”. Do you mean we should use our label selection algorithm to choose target label for existing targeted attacks such as CW-target? If so, we don’t think this experiment is meaningful. In this case, it is meaningless to do subjective experiments because the label for adversarial examples generated by different methods are all the same. It is also meaningless to calculate the attack rate in this case, because the attack rate doesn’t matter to our label selection algorithm, it only matters to the existing targeted attack method itself.
> > >
> > > Or do you mean we should compare our label selection algorithm with some other naïve selections such as random selection? If so, we didn’t compare our selection with random selections because we think it is obvious that our carefully designed label selection algorithm is better than randomly chosen label. We also considered about using the second-highest label as the target label, but we found some drawbacks in doing so. On the one hand, when the network’s confidence score for the highest class is higher than $\delta_1$ (we use 0.8 in this paper), there is no significant difference in confidence scores for other classes, so choosing the second-highest label is not appropriate. On the other hand, when the network’s confidence score for the highest class is lower than $\delta_1$, the second-highest label is usually the ground truth label (In Figure 8 in Appendix D). It will reduce the attack rate if choosing the second-highest label as the target label. So before we read the reviews, we think it is obvious that our carefully designed label selection algorithm is better than second-highest label, too. However, it may be not so obvious. We will add this experiment in our future work.
> > >
> > > Whatever, we appreciate your suggestions, it may be more convincing if we show experimental results to illustrate LabelFool does better than naïve label selections. If you still have questions, don’t hesitate to let us know.

---

> > > > ### Comment · AnonReviewer4 · 2019-11-14
> > > > **Response**
> > > >
> > > > The review system does not provide options between weak reject and weak accept. Thus I cannot change it. I am sorry for that. However, I think AC will care about the opinion.
> > > >
> > > > Sorry for confusing the authors. In comment 3), I meant both of them.
> > > >
> > > > 3-1) Since the paper uses almost one page to introduce the variant of DeepFool, I wanted to see quantitative experiments that show the effectiveness of the proposed targeted attack (Sec. 3.2) over other targeted attack algorithms. I think it is interesting if the proposed targeted attack algorithm outperforms other targeted-attacks when the target labels are less-detectable by humans.
> > > > 3-2) The effectiveness of the proposed label selection algorithm was indeed not obvious. A comparison with the second-highest label will help as the authors mention. I think a comparison with choosing a label with second-largest d(x,y) will also help to prove the effectiveness of the additional complexity in the proposed method. Additionally, analysis of the effect of the hyperparameters \delta_1 and \delta_2 will help to justify the necessity of the division into cases.

---

### Decision · Program_Chairs · 2019-12-19

**Decision:**

Reject

**Comment:**

Thanks for the discussion with reviewers, which improved our understanding of your paper significantly.
However, we concluded that this paper is still premature to be accepted to ICLR2020. We hope that the detailed comments by the reviewers help improve your paper for potential future submission.